# High sensitivity analysis of nanogram quantities of glycosaminoglycans using ToF-SIMS

Andrew L. Hook [1✉], John Hogwood[2], Elaine Gray[2,3], Barbara Mulloy[3] & Catherine L. R. Merry [4]

Glycosaminoglycans (GAGs) are important biopolymers that differ in the sequence of saccharide units and in post polymerisation alterations at various positions, making these complex molecules challenging to analyse. Here we describe an approach that enables small quantities (<200 ng) of over 400 different GAGs to be analysed within a short time frame (3–4 h). Time of flight secondary ion mass spectrometry (ToF-SIMS) together with multivariate analysis is used to analyse the entire set of GAG samples. Resultant spectra are derived from the whole molecules and do not require pre-digestion. All 6 possible GAG types are successfully discriminated, both alone and in the presence of fibronectin. We also distinguish between pharmaceutical grade heparin, derived from different animal species and from different suppliers, to a sensitivity as low as 0.001 wt%. This approach is likely to be highly beneficial in the quality control of GAGs produced for therapeutic applications and for characterising GAGs within biomaterials or from in vitro cell culture.

[1] Advanced Materials and Healthcare Technology, University of Nottingham, Nottingham, UK. [2] National Institute for Biological Standards and Control, Potters Bar, UK. [3] Institute for Pharmaceutical Science, King's College London, Franklin-Wilkins Building, Stamford Street, London, UK. [4] Stem Cell Glycobiology Group, Biodiscovery Institute, Faculty of Medicine and Health Sciences, University of Nottingham, Nottingham, UK. ✉email: andrew.hook@nottingham.ac.uk

Glycosaminoglycans (GAGs) are polysaccharides found within cells, within the pericellular space and as a part of the extracellular matrix (ECM). GAGs regulate biological processes, such as self-renewal, differentiation, growth, inhibition, microbial invasion and defence, with their broad structural diversity and differential localisation accommodating specific interactions with hundreds of binding proteins[1,2]. The complexity of GAGs, including chain length (polymerisation machinery), modification (epimerisation and sulphation of the hydroxyl groups at various positions on the saccharide units) and core protein attachment is orchestrated by enzyme mediated synthesis and allows for GAGs to have greater information carrying capacity than the more commonly studied biological polymers, nucleic acids and proteins.

The five sulfated GAGs, heparin, heparan sulphate (HS), chondroitin sulphate (CS), dermatan sulphate (DS) and keratan sulphate (KS) are synthesised attached to protein cores as proteoglycans, unlike non-sulfated hyaluronan (HA) which is extruded into the pericellular space[3]. Heparin, in the form of a pure polysaccharide released from its core protein, is a globally used anticoagulant and antithrombotic and is currently being considered for anti-inflammatory indications such as chronic obstructive pulmonary disease[4]. Other GAG types are now also increasingly being applied clinically, for example, as treatments for cancer and osteoarthritis, as anti-viral therapies[5] and to support wound healing[6–9]. The rapid and sensitive structural characterisation of GAGs is critical to maintain the standardisation and safety of these animal-derived biomolecules for medical use, as was highlighted by the contamination of heparin samples with over-sulfated CS (OSCS) that led to patient hypotension and death[10]. The ongoing biosecurity of heparin is a significant concern to healthcare systems around the world, necessitating continued efforts to improve heparin analysis and provide synthetic production routes.

Typically, chemical analysis of pharmaceutical GAGs is achieved using nuclear magnetic resonance (NMR) and high performance liquid chromatography (HPLC) methods[11–14]. Simple $^1$H-NMR has been shown to detect 0.1 wt% contaminating OSCS within heparin[15], whilst HPLC achieved a limit of detection of 0.03 wt% for OSCS in heparin and remains the gold standard analysis technique for heparin characterisation[14]. However, some of these approaches require >10 mg of sample, as well as specialised equipment and expert analysis and, therefore, suffer from low throughput[16]. Mass spectrometry plays a leading role in GAG glycomics utilising soft ionisation techniques such as electrospray ionisation[17], however, analysis of whole sulfated GAGs remains difficult[18]. This is particularly problematic for the characterisation of heparin as whole-molecule analysis is necessary to detect inter-species contamination of porcine-derived material used for medical applications[19]. If porcine sources become limited, for example as a consequence of recent outbreaks of African Swine Fever[20], the relatively poor detection of non-porcine material (a limit of detection (LOD) of approximately 2 wt% for detecting a bovine contamination in porcine heparin[21]) is unlikely to be sufficient to protect supplies.

For biomaterial applications requiring surface analysis, X-ray photoelectron spectroscopy has been favoured due to quantitative readouts but is unable to resolve the subtle chemical difference between different GAGs[22]. Time-of-flight secondary ion mass spectrometry (ToF-SIMS) is a promising approach for GAG analysis as spectral acquisition is rapid (≈20 s per sample) and can be applied to whole molecules without the need for purification or enzymatic digestion. ToF-SIMS has been applied to assess the modification of sugars at surfaces but has typically been limited to mono- or di-saccharides[23–28]. Studies of larger polysaccharides, typically heparin or HA, focussed on low mass fragments that

have limited utility to discriminate between the different GAG types[29–35].

In this study ToF-SIMS was used to analyse a microarray containing all six GAG types (analytical preparations of HS, CS, DS, KS, HA, porcine mucosal (PM) heparin, and clinical grade heparin from porcine mucosa, bovine mucosa and bovine lung). Together with principal component analysis (PCA) and partial least square (PLS) regression, this approach was used to chemically distinguish between the different GAG classes in a semi-quantitative manner, whilst notably being able to discern differences between heparin samples derived from different animal sources and different manufacturer batches. The combination of high throughput analysis with high chemical sensitivity indicated the feasibility of this method for quality control of pharmaceutical heparin, detecting possible process related impurities as well as contaminants, and for enabling the surface analysis of GAG-modified materials to facilitate the development of GAG-functional biomaterials.

## Results

**GAG microarray analysis.** Arrays of GAG solutions were prepared using ink-jet printing (Fig. 1a) onto poly-L-lysine (PLL)-coated glass slides, selected for the ability of PLL to adhere GAGs due to ionic interactions, and possible other supramolecular interactions such as hydrogen binding (Supplementary Note 1; Supplementary Figs. 1–2)[36]. Ink-jet printing also enabled the rapid generation of GAG mixtures via in-spot mixing (Supplementary Note 2; Supplementary Figs. 3–4, Supplementary Table 1). Microarrays enable the rapid assessment of libraries of molecules, require small amounts of material (ng) and are compatible with high throughput surface analysis[37]. Microarrays have been widely used to assess DNA, proteins and their analogues (oligonucleotides and peptides)[38–43]. Glycan and GAG microarrays have also been used in alternative applications, but not previously for high-throughput GAG structural analysis[44,45]. In total, approximately 160 ng of material was deposited per spot. Resultant arrays were assessed by bright field microscopy and ToF-SIMS (Fig. 1b, c). All printed spots appeared to be both physically and chemically distinct (Fig. 1c and d). The absence of sulphate signal (SO$_4^-$) for HA samples suggested no carry-over between print runs (Fig. 1c). Regions of interest for each spot were determined from the SO$_4^-$ ion image to enable extraction of spectra for each sample (Fig. 1d, e). A typical spectrum from porcine mucosa (PM) derived heparin exhibited high intensity ions associated with sulphate (SO$_2^-$, SO$_3^-$, C$_3$HSO$_5^+$) and amide (CN$^-$, CNO$^-$) groups as well as highly oxygenated fragments (C$_3$H$_3$O$_2^-$, C$_2$O$_3^+$) (Fig. 1d). Ions associated with the sulphate group were absent from a typical spectrum taken from a HA sample (Fig. 1e).

**Differentiation of GAGs using principal component analysis.** Each sample typically had approximately 900 different ions (both positive and negative). To effectively assess the differences between samples, principal component analysis was used to reduce the dimensionality of the multispectral dataset. Additionally, a sparse dataset was generated to remove uninformative variables not associated with variance between sample types. This is important for the high-dimension dataset where PCA results are difficult to interpret and the sample eigenvectors are not always consistent estimators whilst regression approaches are susceptible to over-fitting[46]. A number of approaches have been used to develop sparsity for PCA[47], including recursive feature selection[48]. In this study, recursive feature addition was used to generate a sparse dataset using the maximisation of the distance between the means of the sample sets as a selection criteria. Recursive feature elimination was then used, using the

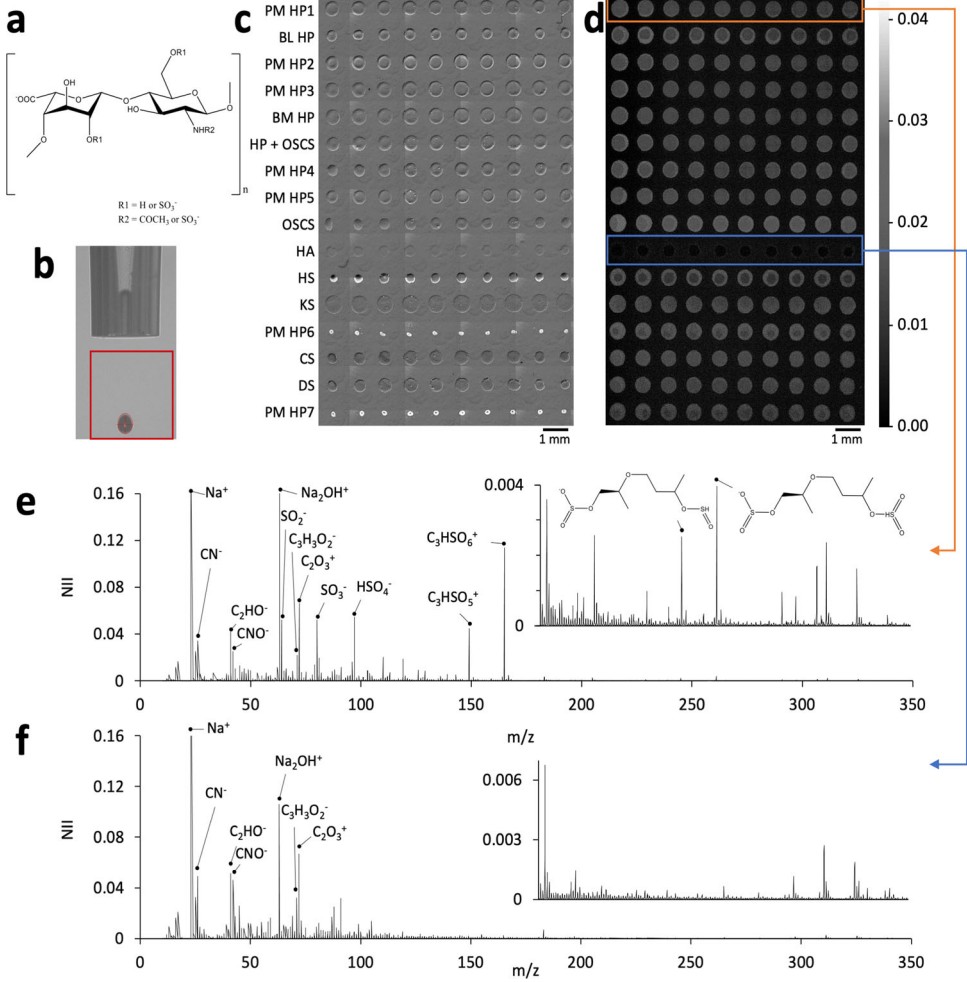

**Fig. 1 Preparation and analysis of a GAG microarray. GAGs were derived from porcine mucosa (PM), bovine mucosa (BM) or bovine lung (BL). If unspecified, samples are derived from PM. a** Chemical structure of the main disaccharide of heparin, in which R1 and R2 are usually $SO_3^-$. **b** Side view image of piezo-dispensing glass nozzle used to dispense GAG solutions to prepare an array. Droplet detection highlighted within the marked region of interest. Typical droplet volume was 320–340 pL. **c** Brightfield microscopy image of prepared array (where print order was sequential from top to bottom) and **d** a corresponding $SO_4^-$ ion image acquired using ToF-SIMS. Intensity scale indicates the measured normalised ion count depicted in **d**. All spots remained distinct and no $SO_2^-$ signal was observed from HA. Region of interest selection for extracting the spectrum from each spot was based upon the high intensity region from the $SO_2^-$ signal, or low intensity area for HA. **e**–**f** Extracted ToF-SIMS spectra for **e** porcine mucosa derived heparin **f** HA.

minimisation of the overlap between 95% confidence ellipses from different sample sets as a selection criterion, to select features that would differentiate between samples with sufficient confidence (Supplementary Figs. 5–6, Supplementary Table 2). To avoid over-fitting, the sample sets were split into training and test sets at a 7:3 ratio (training:test). Test samples were required to fall within the 95% confidence ellipse associated with the principal components describing the variance between samples. The final sparse dataset was further tested for its ability to robustly assess the differences between samples by ensuring sample sets remained separated with multiple randomly generated training/ test sets. Creation of a sparse dataset by this method resulted in 83.5% of the variance captured by PCA to be associated with the difference between the biochemically similar GAGs CS and DS, giving confidence that this approach could also work for a broader set of materials (Supplementary Fig. 7).

The utility of PCA with a sparse dataset to identify differences in GAG samples was assessed for 5 different medical grade PM-derived heparin samples, bovine lung- (BL) and bovine mucosa- (BM) derived heparin, OSCS, CS and a heparin sample contaminated with 1 wt% OSCS from the heparin crisis. A sparse dataset containing 12 different ions was selected. By considering

the scores for PC2 and PC3, the CS samples and contaminated heparin sample were all successfully differentiated from all other heparin samples (Supplementary Fig. 8).

As the ultimate assessment of this approach, samples of each of the 6 GAG types were analysed together to assess whether each sample could be chemically discerned. Without sparsity, PCA was able to successfully differentiate between the KS, HA and OSCS samples, with and without variance scaling (Supplementary Fig. 9a, b). However, separation of the other GAG samples was not achieved, particularly between the different heparin sample sets. The scree plot indicated 1–9 PCs captured variance not associated with noise (Supplementary Fig. 9c). After generation of a sparse dataset, the variance captured by the first 6 principal components (PCs) increased from 78% to 89% due to the removal of features that corresponded to variance not associated with differences between sample set (Supplementary Fig. 9c, d). A total of 48 features were selected for the final sparse dataset that corresponded to the minimum number of features required to produce a high (>0.25) mean average area fraction of ellipses not overlapping.

PCA of the sparse dataset was able to successfully separate all 16 GAG samples to 95% confidence (Figure2). Scores plots for

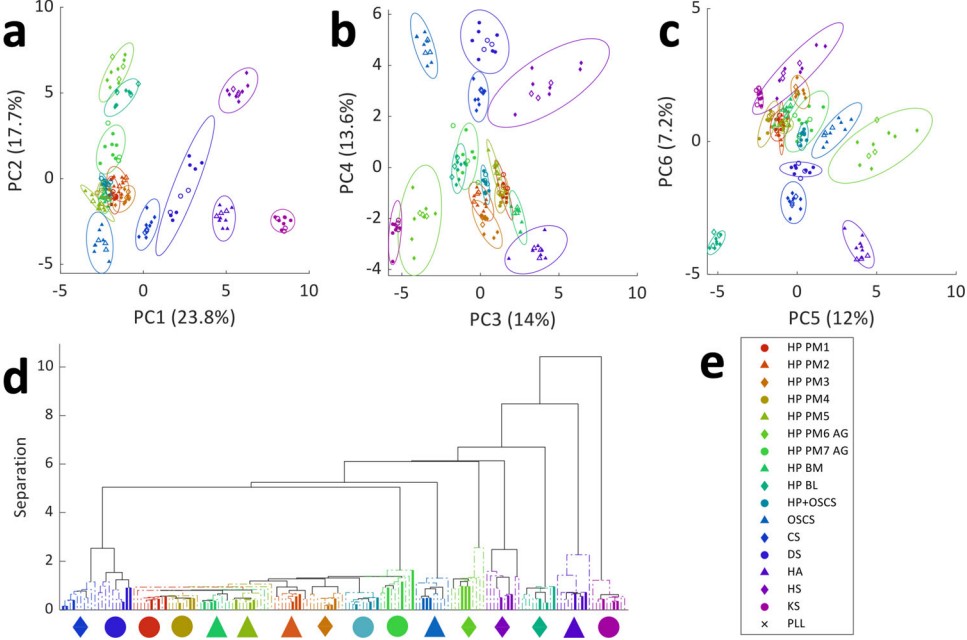

**Fig. 2 PCA of 16 different GAGs. a–c** Scores plots for the 6 first PCs for all 16 samples, including different heparin (HP) samples. Training sets are closed symbols and test sets are open symbols. The 95% confidence ellipse is shown for each sample set. **a** PC1 and PC2, **b** PC3 and PC4, and **c** PC5 and PC6. **d** Dendrogram showing hierarchical clustering of GAG samples based upon the scores for PCs 1–6. Training samples are shown as a dashed line and test samples are shown as a solid line. Lines have been coloured to match sample identity. The associated symbol for each sample type is shown beneath each cluster. **e** Legend showing symbols corresponding to the GAG type.

PCs 1-2 showed clustering of the 6 main types of GAG (Fig. 2a). Further separation of the different types of heparin including separation of heparin from PM, BM or BL and different batches of heparin from PM was achieved by considering PCs 3–6 (Fig. 2b, c). Hierarchical cluster analysis was used to classify the different samples based upon their Euclidean distance. The outcome of this unsupervised classification approach is shown as a dendrogram (Fig. 2d), where samples that are most similar are positioned together. In all cases, samples were clustered within their correct sample group, including the test set, with the exception of single replicates of two heparin PM batches and one replicate of the heparin BM sample.

Only those ions with a possible assignment based upon the elemental composition of GAGs were selected (C, O, H, N, S). Each ion was assigned a loading for each PC, shown in Supplementary Table 3. Possible assignments for each of the 48 ions for the key PCs is listed in Supplementary Table 3. A number of ions likely associated with sulphate groups were selected, including ions $SN^+$ and $SNO_2^-$, as well as larger ions such as $C_{10}H_{11}SO_4^-$. This suggests that part of the variance captured by the PCA was associated with the sulfation patterns on the GAGs. Ions likely associated with di- and tri- saccharides, such as $C_{18}H_{33}SO_5^+$, $C_{18}H_{38}O_9^+$, were also selected. Further interpretation of the relation between the ions identified and the GAG structures is limited due to the relatively low mass resolution of the ToF mass analyser.

To test the capability of ToF-SIMS analysis to chemically distinguish between samples in a more complex biological environment, each of the 6 GAG types were added to a fibronectin (FN) solution, printed as a microarray and analysed by ToF-SIMS. FN is a common component of biological ECMs as well as serum and plasma. After generating a sparse dataset, the multispectral data was assessed for its ability to distinguish between the different samples using PCA and hierarchical cluster analysis. All 6 GAG types were chemically differentiated from each other, and from pure FN (Fig. 3a), where all samples,

including the test sets, were successfully categorised using hierarchical cluster analysis (Fig. 3a). Possible assignments for the 18 ions selected for this sparse dataset and their loadings are shown (Supplementary Table 4). Similar to the model without FN, ions containing sulphate groups ($CHSO^-$, $C_3HSO_2^-$) were present in the model. Most of the ions selected were small in nature and likely derived from a monosaccharide. This may be due to a reduced yield of higher molecular weight ions associated with GAGs from within a protein matrix. Additionally, ions likely associated with FN ($CH_4N^+$) were also selected.

**Quantification of spiked samples using partial least square regression**. To assess the sensitivity of the analysis methodology to adulteration, a PM heparin was spiked with increasing concentrations of either OSCS, BM heparin or BL heparin. The ToF-SIMS spectral data was correlated with the fraction of spiking agent using PLS regression, as has been done previously for correlating water contact angle or protein adsorption with ToF-SIMS data[49,50]. Initially, a sparse dataset was selected for each sample set by least absolute shrinkage and selection operator (LASSO) to minimise over-fitting by removal of uninformative features[51]. The number of latent variables used was selected based upon the minimisation of the root mean square error of cross validation (Supplementary Fig. 11). Plots of the measured fractions of spiking agent and those predicted from the ToF-SIMS data using the PLS model are shown in Supplementary Fig. 12. A high correlation ($R^2 > 0.94$) between measured and predicted values was observed for samples spiked with either OSCS or heparin BM, suggesting that the ToF-SIMS data was able to distinguish differences in samples down to 0.001 wt%. This was confirmed by PCA of the same samples, which demonstrated separation between non-spiked samples and the samples spiked at 0.001 wt% to 95% confidence (Supplementary Fig. 13, Supplementary Tables 5–7). A weaker correlation ($R^2 = 0.88$) was observed for the samples spiked with BL heparin (Supplementary Fig. 12). PCA of these samples showed that the non-spiked

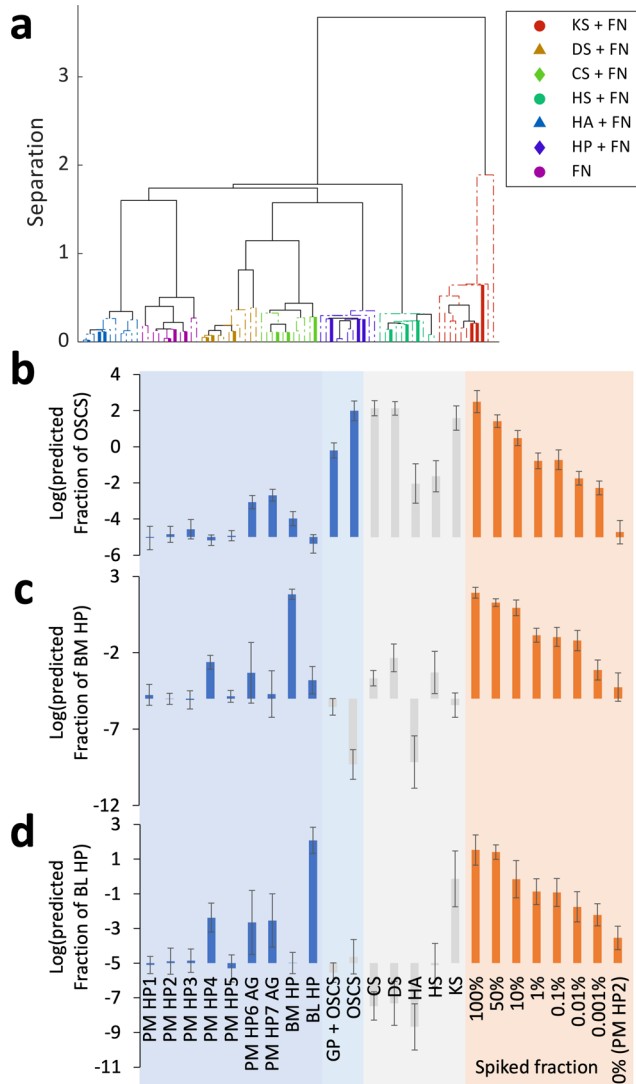

**Fig. 3 Summary of multivariate analysis of mixed GAG samples. a** PCA of datasets from 6 GAG types mixed with FN. Dendrogram showing hierarchical clustering of GAG samples based upon the scores for PCs 1 and 2 (Supplementary Fig. 10). Training samples are shown as a dashed line and test samples are shown as a solid line. Lines have been coloured to match the sample identity. Legend of sample identity shown with figure. **b–d** Predicted values for the heparin/OSCS samples measured as a part of the 16 GAGs (■), unrelated GAG types (■) and heparin (HP) PM spiked with **b** OSCS, **c** HP BM and **d** HP BL (■) using the corresponding PLS models. Error bars show ± 1 standard deviation unit, $n = 10$.

sample and the 0.001 wt% sample could not be separated to 95% confidence (Supplementary Fig. 13b). However, a linear response between the measured and predicted fraction of BL heparin was observed down to 0.01 wt%, suggesting that the analysis was sensitive to this concentration. Similar $R^2$ values were observed for the training (70%) and test (30%) sets for all models, suggesting there was no over-fitting.

The PLS models were used to predict the fraction of the contaminant in each of the different heparin samples initially assessed by PCA (Fig. 3b–d). The amount of OSCS predicted in all heparin samples was below 0.001%, with the exception of the analytical grade heparins, which had predicted OSCS fractions of 0.0009 and 0.002 wt%. High predicted fractions (≈100 wt%) were predicted for the OSCS sample, whilst the sample with a known

OSCS adulteration of 1 wt% had a predicted OSCS fraction of 0.6 wt%. Quantitative analysis of samples using ToF-SIMS data is limited by matrix effects[52]. Therefore, the PLS model was only applicable to samples analysed within the same matrix environment as the training data. For predictions of the bovine-derived heparin content, low values ($1 \times 10^{-5}$ wt%) were predicted for the porcine-derived heparin samples with the exception of the analytical grade samples and sample 4, which all had predicted values of approximately $1 \times 10^{-3}$ wt%. The lower purity of the analytical grade heparin is expected, and our results indicate low levels of OSCS contamination. The presence of bovine-derived heparin in sample 4 was unexpected, but does coincide with lower levels of anticoagulant activity observed for this sample (Supplementary Table 8).

For completion, the PLS models were also applied to unrelated GAG types, shown in Fig. 3b–d. Although it is possible to suggest that high levels of OSCS were predicted in the DS and CS samples, the predicted values are unreliable as the models were trained for the detection of specific contaminants in heparin. The successful separation of each of the 16 GAGs by PCA suggests that, in principle, PLS models of pairwise mixtures of the other GAG types could be prepared.

Each of the features selected for the PLS regression was assigned a regression coefficient (RC) that informed how strongly it influenced the model and whether it was associated with the contaminant or PM heparin. Tables of possible assignments for the ions selected for each model and their associated RCs are shown in Supplementary Tables 9–11. For each model, ions likely derived from mono- and di-saccharides were associated with PM heparin (having a negative RC) including $C_{13}H_{29}S_2NO_4^+$, $C_{16}H_{29}N_2O_7^-$ and $C_{32}H_{56}N_3O_{12}^-$. Furthermore, ions containing sulphate groups, such as $CH_4SNO_2^-$, $C_5HSNO_6^-$ and $C_2SNO_3^-$ were also selected, suggesting the model includes information both about the disaccharide sequence of the heparin molecules and the sulfation pattern. The ions associated with the spiked GAGs (OSCS, BM heparin and BL heparin) included ions likely representative of the disaccharide sequence ($C_{20}H_{37}SN_2O_5^-$, $C_{12}H_{29}N_2O_7^+$ and $C_{22}H_{43}N_2O_7^+$) or sulfation pattern ($KC_5SNO^-$, $C_3H_5SNO_3^-$, $C_2H_3S^+$) of the spiked GAGs. There is large uncertainty regarding the ion assignments, particularly for large ions, due to the mass resolution of the ToF analyser. The suggestions provided are based upon structures that match GAG stoichiometry and have a minimal deviation between the measured and theoretical values.

**Assessment of heparin activity**. The anticoagulant action of heparin is chiefly due to its ability to potentiate the serine protease inhibitor antithrombin, a protein normally present in plasma. Assays of antithrombin mediated inhibition of the clotting factors thrombin (factor IIa) and of factor Xa, using purified proteins, are used to determine the potency of clinical grade heparin in International Units (IU)/mg. The Activated Partial Thromboplastin Time (APTT) is a plasma-based method for measuring anticoagulant activity.

The specific activities of five heparin samples were measured by these three methods and the results are summarised in Supplementary Table 5. A number of ions were found to significantly ($p < 0.001$) correlate linearly (Pearson's $r > 0.75$) with each of the measures of activity, as shown in Supplementary Fig. 14. The origin of these ions is not known but the correlations suggest that they arise from structural factors that determine anticoagulant activity, either very specifically in terms of the rare pentasaccharide motif that determines affinity for antithrombin[53], or in more general terms such as overall degree of sulfation. This link between the surface chemistry as measured

by ToF-SIMS and a quantitative measure of biological activity is unexpected.

## Discussion

Analytical characterisation of GAGs underpins multiple aspects of current GAG-related research, including the understanding of their fundamental biological roles. GAGs are already important pharmaceutical compounds (as discussed above for heparin) and are increasingly being used for various therapeutic applications as well as being incorporated into biomaterials for improved biofunctionality[54–57]. Mass spectrometry techniques focus on the analysis of oligosaccharides for the purposes of sequencing[58,59]. The use of ToF-SIMS to analyse GAG samples on an arrayed platform provides a methodology by which small quantities (< 200 ng) of hundreds of different GAGs could be analysed within a short time window (3–4 h). The resultant spectra were derived from the whole molecules and did not require any pre-digestion or pre-labelling of material. The analysis was informative of the GAG disaccharide sequence, sulfation pattern and biological activity and enabled discernment between all 6 different GAG types investigated.

The high throughput and sensitivity achievable by this system is also important for the quality control of GAGs within healthcare settings to ensure patient safety. For the most widely used GAG in medicine, heparin, recent problems in pharmacovigilance have been the spur to develop a battery of orthogonal tests to ensure identity, purity and high specific bioactivity[12]. Besides the detection of contaminants, whether introduced accidentally or as deliberate adulteration, it is necessary to monitor impurities in heparin that can arise both from co-purification of related compounds such as chondroitin and dermatan sulphates and from minor chemical modifications arising in the manufacturing process[60,61]. Whole molecule analysis is desirable to be able to detect contaminants and process-related impurities in active pharmaceutical ingredient of GAG-based products, for example, detection of mixed-species heparin. Whilst whole molecule analysis of GAGs has been achieved[62], the approaches are typically slow, require large amounts of samples and lack sensitivity. We applied our protocol to pharmaceutical grade heparin derived from different animal species and from different suppliers. Our approach allowed for the clear identification of heparin samples in terms of species of origin, and highly sensitive detection of contaminants spiked into PM heparin, including a sensitivity of 0.001 wt% of the addition of OSCS, the contaminant associated with the heparin crisis, and to 0.01 wt% for BL heparin in PM heparin. This approach is likely to be highly beneficial in the quality control of GAGs produced for therapeutic applications and for characterising GAGs within biomaterial systems or from in vitro cell culture.

The use of multivariate analysis approaches was necessary to interrogate the multi-dimensional ToF-SIMS datasets. PCA has been widely used to assess the variance within ToF-SIMS datasets and was used here to be able to capture the variance between different GAG samples, whilst PLS regression demonstrated that the fraction of a spiked GAG could be predicted from the ToF-SIMS spectra. Creation of sparse datasets was important to avoid over-fitting data as well as to remove uninformative features. For PCA, recursive feature selection identified the ions that captured the variance between the different GAGs including within a more complex biological environment containing fibronectin.

Implementation of the approach described as either a tool for basic research or as a quality control methodology for heparin manufacturer would require for the data readouts to be reached without intervention of expert users. The data models established in this study would provide a useful system that future samples could be applied to, with the possibility to identify unknown GAGs (PCA) or detect contamination within a sample (PLS). Unsupervised approaches like hierarchical cluster analysis provide a mechanism by which useful readouts can be obtained without any user intervention. The models can also easily be further expanded and made more robust through the addition of further control samples, whilst models focussing on a single GAG type are also easily achievable. The approach therefore, has broad applicability and can be readily adapted to various GAG-based applications.

## Methods

**Materials**. HS Na salt from porcine mucosa (Iduron), DS Na salt from porcine mucosa (Average Mw = 41,000, Iduron), CS B Na salt from porcine mucosa (Sigma-Aldrich), HA Na salt from *Streptococcus equi* (Mw = 15,000–30,000, Sigma-Aldrich), heparin from porcine mucosa (Mw = 5000, Fisher Scientific) were used as received. GAGs were prepared as standard solutions of 5 mg/ml in ultra-pure water (Purelab Ultra, ELGA LabWater). Heparin samples received from the NIBSC heparin archive. KS Na salt was derived from bovine corneal. Fibronectin was derived from bovine plasma (Sigma-Aldrich lot#101M7012V). Poly-L-lysine coated slides (Poly-Prep, Sigma-Aldrich), aminoalkylsilane functionalised slides (Silane-Prep, Sigma-Aldrich), tissue culture polystyrene (TCPS, Nunclon Delta, ThermoFisher Scientific), allylamine plasma polymer coated polystyrene (EpranEx, BD Biosciences) and bare glass slides (Corning) were used as received.

**Microarray preparation**. Arrays were prepared using an s11 sciFLEXARRAYER dispensing system (Scienion) using a glass piezo dispense capillary (P-2020, Scienion). Drop volumes were ≈ 300 pL, as measured using the drop shape analyser tool (Scienion) prior to each run. Print runs were conducted at a relative humidity of 65 % at room temperature. GAG solutions were diluted to 2–5 mg/ml in a polypropylene 384-well plate (Corstar) in ultra-pure water (18.2 MΩ.cm) with or without 1 mg/ml fibronectin. For in-spot mixing, 0–150 nL of water was printed and subsequently GAGs were dosed into the water droplets to facilitate mixing prior to surface adsorption. The nozzle was flushed with 250 μL of water whilst the outside of the nozzle was washed with copious amounts of water between printing different samples.

**ToF-SIMS analysis**. Time-of-flight secondary ion mass spectrometry measurements were conducted using a ToF-SIMS IV (IONTOF GmbH, Münster, Germany) instrument operated using a 25 keV $Bi_3^+$ primary ion source exhibiting a pulsed target current of >0.3 pA. Samples were scanned at a pixel density of 512 pixels per mm, with fifteen shots per pixel over a given area. An ion dose of $2.45 \times 10^1$ ions per $cm^2$ was applied to each sample area ensuring that static conditions were maintained throughout. Both positive and negative ion spectra were collected (mass resolution of >7000 at $m/z = 29$). Owing to the non-conductive nature of the samples, charge compensation was applied in the form of a low energy (20 eV) electron floodgun. Patch areas of $0.5 \times 0.5$ mm were acquired at a resolution of $256 \times 256$ pixels by rastering the primary ion beam over the patch using a 'random raster' path sequence. Patch areas were sequentially acquired over the entire microarray using programmed stage movements through the macro-raster stage function. The patch areas were combined into a mosaic image, allowing all patches to be processed together. A peak list was produced using the peak search tool (SurfaceLab 6, IONTOF), minimum counts set to 100, maximum background set to 0.8. To ensure the peak search tool had successfully identified peaks, all ions of interest were visually inspected. Regions associated with each polymer spot were then extracted and recalibrated, and the peak list was applied to produce an individual spectrum for each polymer. In total, 412 positive and 460 negative ion peaks were identified. Peak assignments were achieved using a custom built Visual Basic Application algorithm (PeakAssigner v2.6)[63]. Only peaks with a chemical assignment derived from C, S, O, N and H within 100 ppm were used for PCA.

**Light microscopy**. Phase contrast microscopy images were acquired using an Olympus IX51 microscope using a 40× objective, NA = 0.13. The microscope was equipped with a Smart Imaging System (IMSTAR) using Fluo/LightVision software (v6.04 K).

**Principal component analysis**. A microarray of samples was initially prepared to enable a large number of samples to be rapidly assessed. The arrays were analysed by ToF-SIMS and spectra were obtained for each sample. Datasets were variance scaled and mean-centred and replicate measurements were split into training (70%) and test (30%) sets.

Principal component analysis (PCA) was conducted using the function 'pca' within Matlab R2018a (9.4.0.813654) on the full dataset. The scree plots were used to identify the total number of latent variables associated with meaningful variance by fitting a linear curve to high latent variable values (typically 15–20) and observing where the variance explained departed from linearity for lower numbers

of latent variables. A sparse dataset was then created by recursive feature addition using the minimisation of confidence ellipse overlap as a weighting from the scores plots of relevant latent variables. After each feature addition the resulting models were checked to ensure the test datasets fell within the 95% confidence limits of the training datasets. The minimum number of variables required to achieve the minimum amount of ellipse overlap was selected. PCA was conducted on the final sparse dataset to assess differences between samples. The function 'linkage' within Matlab 2018a (9.4.0.813654) was used for hierarchical cluster analysis using the scores of relevant latent variables from the final PCA.

All data processing was conducted using a custom-built Matlab project.

**Partial least square regression**. Partial least square regression was conducted in Matlab R2018a (9.4.0.813654) using the plsregress function, that utilises the SIMPLS algorithm. The ToF-SIMS spectra, variance scaled and mean-centred prior to analysis, was used as a set of predictors whilst the concentration of spiked agent was used as the response. Replicate measurements were split into training (70%) and test (30%) sets. Sparse datasets were produced by LASSO using the lassoglm function, with the lambda value selected based upon the minimisation of the standard error. The final PLS models used the sparse datasets, and the number of latent variables was selected as the minimisation of the root mean square error of cross-validation.

## Data availability

Data used to produce figures is available at the University of Nottingham data repository https://doi.org/10.17639/nott.7105. The Matlab project used for data processing is available at github https://github.com/fishhooky/PCABundle.

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

## Acknowledgements

Andrew Hook kindly acknowledges the University of Nottingham for provision of his Nottingham Research Fellowship. Catherine Merry kindly acknowledges support from the National Centre for the Replacement Refinement and Reduction of Animals in Research (NC/T2T0219). Prof. Ulf Lindahl is kindly acknowledged for the provision of keratan sulphate samples.

## Author contributions

A.L.H. and C.L.R.M. conceived the study. A.L.H. acquired funding, developed the methodology, performed experiments and analysed the data. JH performed heparin activity studies. A.L.H., J.H., E.G., B.M. and C.L.R.M. contributed to the writing of the paper.

## Competing interests

A.L.H. and C.L.R.M. declare the following competing interest: Patent application 2104133.0. Inventors: A.L.H. and C.L.R.M. Applicant: University of Nottingham. The remaining authors declare no competing interests.
