## [Peer Review File · Communications Chemistry]

Reviewers' comments:

Reviewer #1 (Remarks to the Author):

In this article, a TOF-SIMS method has been developed to analyze and differentiate GAGs. The method exhibited superior sensitivity and was capable to distinguish differences in samples down to 0.001 wt%. However, the authors only showed that the different GAGs generated differentiable TOF-SIMS spectra and PCA, but failed to explain the connection between TOF-SIMS spectra and the structure of GAGs. The disaccharide composition, the sulfation pattern, the uronate epimerization, or the molecular weight, which determined or impacted on the TOF-SIMS spectra, and how? A clear mechanism will be appreciated.

Reviewer #2 (Remarks to the Author):

The manuscript "High sensitivity analysis of nanogram quantities of glycosaminoglycans using ToF-SIMS" by Hook et al. reports an exciting new technique for the analysis of heparin and related polysaccharides using a combination of mass spectrometry and chemometric tools. This is a very significant development in the field, which for decades lacked reliable and comprehensive analytical tools to assess the quality of pharmaceutical GAG-based products, as well as to provide meaningful characterization of clinical/biological GAG samples. The approach presented in the manuscript bypasses the tedious step of data interpretation (which always requires an expert to be involved, inevitably leading to the escalation of the cost and the increase of the analysis time), and instead relies on chemometrics (principal component analysis) to not only make a clear distinction among the different GAG species, but also to differentiate among the samples of the same GAG species procured from different sources. In fact, the method is so robust and sensitive that it allows low-level impurities to be readily detected. Undoubtedly, this is a truly remarkable development in the field of GAG analysis, which is going to close one of the most significant (and enduring) gaps. The only factor that may prevent the immediate adoption of this technique by the GAG biologists and the pharma community is the relatively high cost of the ToF-SIMS equipment (and the fact that they are still not a common tool in the instrumental arsenal of bioanalytical labs).

The paper is well-written and succinct, and follows a clear logical outline. All conclusions are strongly supported by the data presented in the manuscript (and the vast array of the data in the Supplementary Material section). The manuscript is already in an excellent shape, and in my opinion can be published in its present form. However, should the revision be deemed necessary by the Editor, the authors are encouraged to consider the following minor points:

1) p. 1, line 14: "ion" is missing between "secondary" and "mass spectrometry"

2) p. 1, line 22: the Introduction section begins with a statement that "Glycosaminoglycans (GAGs) are polysaccharides found at the interface BETWEEN cells and the extracellular matrix," but aren't most GAGs a part of the ECM? Also, it would probably be prudent to expand this sentence to reflect the fact that many GAGs are localized INSIDE the cells (e.g., heparin in the granules of the mast cells, CS in the platelets' alpha-granules, etc.).

3) p. 3, line 120: the authors state that "The scree plot indicated 1-9 PCs captured variance not associated with noise (Figure SI9c)," but what the plot actually shows is the monotonic decrease of the variance capture for each latent variable. What was the criterion employed for selecting the cut-off value?

Overall, this is an excellent manuscript, which certainly merits publication in Chem. Comm.

Reviewer #3 (Remarks to the Author):

Manuscript authored by Hook and colleagues describes a MS-based method to analyze glycosaminoglycans. Authors claim that the analysis can be achieved without subjecting to the depolymerization step, which potentially simplifies and shortens the analysis. Authors further indicate that this method can be used to distinguish heparin isolated from different sources. If the method works, it is important for a wide range of application.

Although the study appears important, this reviewer is frustrated by the presentation. It is very hard to understand what authors did in the experiment and how to do the data analysis. The supplementary part is excessively wordy and long, very dense to read. Authors should make extra efforts to communicate the research better.

Here are some specific comments:

1. Why such studies must be carried out on a microarray chip or involved the use array printer?
2. What is the molecular mechanism to distinguish porcine heparin and bovine heparin under TOF-SIMS conditions?
3. How do the authors achieve quantitative analysis of heparin or glycosaminoglycan samples?
4. One critical issue in the analysis of GAG samples isolated from biological sources is low ionization efficiency due to contaminants in the preparation. Here, they used highly purified GAG to complete the analysis. Did authors take consideration of the impact of proteins or different salt contaminants on the analysis?

Reviewer #1 (Remarks to the Author):

In this article, a TOF-SIMS method has been developed to analyze and differentiate GAGs. The method exhibited superior sensitivity and was capable to distinguish differences in samples down to 0.001 wt%. However, the authors only showed that the different GAGs generated differentiable TOF-SIMS spectra and PCA, but failed to explain the connection between TOF-SIMS spectra and the structure of GAGs. The disaccharide composition, the sulfation pattern, the uronate epimerization, or the molecular weight, which determined or impacted on the TOF-SIMS spectra, and how? A clear mechanism will be appreciated.

The relationship between the differences in the ToF-SIMS spectra of the different GAGs and the GAG structure is indeed of interest and we thank the reviewer for raising this important point. In the paper we show possible assignments for the SIMS ions associated with the key differences and have noted in the manuscript that these are likely associated with differences in sulfation pattern and uronate epimerisation. However, the relatively poor mass resolution of the ToF mass analyser makes it difficult to confidently assign the ions and limits the extent to which these ions can be used to provide insight into GAG structure. We have been careful to not overstate our findings.

Enabling interpretation of the SIMS ions identified in light of GAG structure would require analysis using a higher mass resolution mass analyser and the study of appropriate reference materials. This is the subject of on going work, but is beyond the scope of this present study. In the meantime, analytical use of this technique to distinguish between closely related GAG structures and mixtures of GAGs with high sensitivity does not depend on full structural interpretation of the data. Indeed, this is a recognised advantage of chemometric methods.

We have acknowledged this limitation with the addition of the following sentences.

'Further interpretation of the relation between the ions identified and the GAG structures is limited due to the relatively low mass resolution of the ToF mass analyser. Analytical use of this technique to distinguish between closely related GAG structures and mixtures of GAGs with high sensitivity does not depend on full structural interpretation of the data. Indeed, this is a recognised advantage of chemometric methods.'

Reviewer #2 (Remarks to the Author):

The manuscript "High sensitivity analysis of nanogram quantities of glycosaminoglycans using ToF-SIMS" by Hook et al. reports an exciting new technique for the analysis of heparin and related polysaccharides using a combination of mass spectrometry and chemometric tools. This is a very significant development in the field, which for decades lacked reliable and comprehensive analytical tools to assess the quality of pharmaceutical GAG-based products, as well as to provide meaningful characterization of clinical/biological GAG samples. The approach presented in the manuscript bypasses the tedious step of data interpretation (which always requires an expert to be involved, inevitably leading to the escalation of the cost and the increase of the analysis time), and instead relies on chemometrics (principal component analysis) to not only make a clear distinction among the different GAG species, but also to differentiate among the samples of the same GAG species procured from different sources. In fact, the method is so robust and sensitive that it allows low-level impurities to be readily detected. Undoubtedly, this is a truly remarkable development in the field of GAG analysis, which is going to close one of the most significant (and enduring) gaps. The only factor that may prevent the immediate adoption of this technique by the GAG biologists and the pharma community is the

relatively high cost of the ToF-SIMS equipment (and the fact that they are still not a common tool in the instrumental arsenal of bioanalytical labs).

The paper is well-written and succinct, and follows a clear logical outline. All conclusions are strongly supported by the data presented in the manuscript (and the vast array of the data in the Supplementary Material section).

We thank the reviewer for their supportive comments.

The manuscript is already in an excellent shape, and in my opinion can be published in its present form. However, should the revision be deemed necessary by the Editor, the authors are encouraged to consider the following minor points:

1) p. 1, line 14: "ion" is missing between "secondary" and "mass spectrometry"

We thank the reviewer for identifying this error. We have corrected as suggested.

2) p. 1, line 22: the Introduction section begins with a statement that "Glycosaminoglycans (GAGs) are polysaccharides found at the interface BETWEEN cells and the extracellular matrix," but aren't most GAGs a part of the ECM? Also, it would probably be prudent to expand this sentence to reflect the fact that many GAGs are localized INSIDE the cells (e.g., heparin in the granules of the mast cells, CS in the platelets' alpha-granules, etc.).

We have altered the first sentence as follows:

'Glycosaminoglycans (GAGs) are polysaccharides found within cells, within the pericellular space and as a part of the extracellular matrix (ECM).'

3) p. 3, line 120: the authors state that "The scree plot indicated 1-9 PCs captured variance not associated with noise (Figure S19c)," but what the plot actually shows is the monotonic decrease of the variance capture for each latent variable. What was the criterion employed for selecting the cut-off value?

We have added the following text to Figure S19 heading to explain the selection of latent variables.

'Selection of number of latent variables used was determined by fitting a linear curve to the variance explained curve for high numbers of latent variables (15-20) and selecting where the variance explained departed from linearity for reduced numbers of latent variables.'

We have also added the following sentence to the methods.

'The scree plots were used to identify the total number of latent variables associated with meaningful variance by fitting a linear curve to high latent variable values (typically 15-20) and observing where the variance explained departed from linearity for lower numbers of latent variables.'

Overall, this is an excellent manuscript, which certainly merits publication in Chem. Comm.

Reviewer #3 (Remarks to the Author):

Manuscript authored by Hook and colleagues describes a MS-based method to analyze glycosaminoglycans. Authors claim that the analysis can be achieved without subjecting to the depolymerization step, which potentially simplifies and shortens the analysis. Authors further indicate that this method can be used to distinguish heparin isolated from different sources. If the method works, it is important for a wide range of application.

Although the study appears important, this reviewer is frustrated by the presentation. It is very hard to understand what authors did in the experiment and how to do the data analysis. The supplementary part is excessively wordy and long, very dense to read. Authors should make extra efforts to communicate the research better.

We have clarified the methodology used by splitting the multivariate analysis section up into separate PCA and PLS sections and have added further details to ensure that the methods used are presented clearly. We have also revised the results discussion within the supplementary information to ensure the text is succinct and clear. We agree that the supplementary information is indeed long and information dense, however, as we are presenting a new method it is important to provide justification for the various decisions that have been made in regard to the analysis. This discussion is intended for expert users and will likely not be as accessible to a general audience. We have deliberately chosen to move the technical aspects of the method development to the supplementary information to ensure the main manuscript is clear, succinct and accessible for a non-specialist audience.

The altered methods sections now read as follows:

1.1. Principal component analysis

A microarray of samples was initially prepared to enable a large number of samples to be rapidly assessed. The arrays were analysed by ToF-SIMS and spectra were obtained for each sample. Datasets were variance scaled and mean-centred and replicate measurements were split into training (70%) and test (30%) sets.

Principal component analysis (PCA) was conducted using the function 'pca' within Matlab R2018a (9.4.0.813654) on the full dataset. The scree plots were used to identify the total number of latent variables associated with meaningful variance by fitting a linear curve to high latent variable values (typically 15-20) and observing where the variance explained departed from linearity for lower numbers of latent variables. A sparse dataset was then created in two steps. First recursive feature addition was used to add ions to the dataset that maximised the Euclidian distance between the means of the sample sets. This was continued until the addition of further features provided no further increase. Recursive feature elimination was then used to reduce the dataset using the minimisation of confidence ellipse overlap from the scores plots of relevant latent variables as a weighting. After each feature addition the resulting models were checked to ensure the test datasets fell within the 95% confidence limits of the training datasets. The minimum number of variables required to achieve the minimum amount of ellipse overlap was selected. PCA was conducted on the final sparse dataset to assess differences between samples using the number of latent variables indicated by the scree plot generated using the sparse dataset. The function 'linkage' within Matlab 2018a (9.4.0.813654) was used for hierarchical cluster analysis using the scores of relevant latent variables from the final PCA.

All data processing was conducted using a custom-built Matlab project⁶⁴.

1.2. Partial least square regression

Partial least square regression was conducted in Matlab R2018a (9.4.0.813654) using the plsregress function, that utilises the SIMPLS algorithm. The ToF-SIMS spectra, variance scaled and mean-centred prior to analysis, was used as a set of predictors whilst the log of the concentration of spiked agent was used as the response. Replicate measurements were split into training (70%) and test

(30%) sets. Sparse datasets were produced by LASSO using the lassoglm function, with the lambda value selected based upon the minimisation of the standard error. The final PLS models used the sparse datasets, and the number of latent variables was selected as the minimisation of the root mean square error of cross-validation. The models could then be used to predict the log of the concentration of spiked agent for unknown samples.

Here are some specific comments:

1. Why such studies must be carried out on a microarray chip or involved the use array printer?

The study does not have to be carried at as a microarray, but it offers the advantages listed in the following sentence in the manuscript:

'Ink-jet printing also enabled the rapid generation of GAG mixtures via in-spot mixing (Figure SI3-4). Microarrays enable the rapid assessment of libraries of molecules, require small amounts of material (ng) and are compatible with high throughput surface analysis.'

2. What is the molecular mechanism to distinguish porcine heparin and bovine heparin under TOF-SIMS conditions?

As discussed in the response to reviewer 1, this relationship, although of interest, is beyond the scope of the current study.

3. How do the authors achieve quantitative analysis of heparin or glycosaminoglycan samples?

The quantitative analysis of heparin is indeed limited by the semi-quantitative nature of SIMS and possible matrix effects. In order to achieve quantitative analysis we prepared reference samples to observe the response of SIMS ions in the varied matrix environments. As a result we are able to observe the trends of different ions across the different matrix environments and identify how these ions can be interpreted in order to give a quantitative measurement. This is achieved through the construction of a PLS model and selection of appropriate ions.

We have added the following text to highlight the important impact of matrix effects on our results.

'Quantitative analysis of SIMS data is limited by matrix effects. Therefore, the PLS model is only applicable to samples analysed within the same matrix environment as the training data.'

4. One critical issue in the analysis of GAG samples isolated from biological sources is low ionization efficiency due to contaminants in the preparation. Here, they used highly purified GAG to complete the analysis. Did authors take consideration of the impact of proteins or different salt contaminants on the analysis?

As shown in Figure 3a, we studied the ability of the approach to work in a high protein environment. Here, we were able to see that the different GAGs could still be differentiated. The quantitative analysis was focussed on heparin samples intended for use as a pharmaceutical product. These samples are highly purified and for this application it is highly relevant to be studying these sorts of samples.

REVIEWERS' COMMENTS:

Reviewer #3 (Remarks to the Author):

The revised manuscript has been significantly improved. This is indeed a piece of very innovative and important work for heparin and related polysaccharide research. There are no further questions from this reviewer.